# Evaluation of a Novel Mitochondrial Pan-*Mucorales* Marker for the Detection, Identification, Quantification, and Growth Stage Determination of Mucormycetes

**DOI:** 10.3390/jof5040098

**Published:** 2019-10-11

**Authors:** Rita Caramalho, Lisa Madl, Katharina Rosam, Günter Rambach, Cornelia Speth, Johannes Pallua, Thomas Larentis, Ricardo Araujo, Ana Alastruey-Izquierdo, Cornelia Lass-Flörl, Michaela Lackner

**Affiliations:** 1Institute of Hygiene and Medical Microbiology, Medical University of Innsbruck, Schöpfstraße 41, 6020 Innsbruck, Austria; rita.dinora@hotmail.com (R.C.); lisa.madl@hotmail.com (L.M.); katharina.rosam@i-med.ac.at (K.R.); Guenter.Rambach@i-med.ac.at (G.R.); Cornelia.Speth@i-med.ac.at (C.S.); thomas.larentis@rolmail.net (T.L.); Cornelia.lass-floerl@i-med.ac.at (C.L.-F.); 2Institute of Pathology, Neuropathology and Molecular Pathology, Medical University of Innsbruck, Müllerstraße 44, 6020 Innsbruck, Austria; Johannes.Pallua@i-med.ac.at; 3i3 S, Instituto de Investigacao e Inovacao da Universidade do Porto, R. Alfredo Allen, 4200-135 Porto, Portugal; ricjparaujo@yahoo.com; 4Department of Medical Biotechnology, School of Health Sciences, Flinders University, Sturt Rd, Bedford Park SA 5042, Australia; 5Mycology Reference Laboratory, National Centre for Microbiology. Instituto de Salud Carlos III. Majadahonda, 28220 Madrid, Spain; anaalastruey@isciii.es

**Keywords:** mucormycosis, HRM, qPCR, large subunit ribosomal DNA, mitochondrial genes, growth stage, activity marker, seminested PCR, pan-*Mucorales*

## Abstract

Mucormycosis infections are infrequent yet aggressive and serious fungal infections. Early diagnosis of mucormycosis and its discrimination from other fungal infections is required for targeted treatment and more favorable patient outcomes. The majority of the molecular assays use 18 S rDNA. In the current study, we aimed to explore the potential of the mitochondrial rnl (encoding for large-subunit-ribosomal-RNA) gene as a novel molecular marker suitable for research and diagnostics. Rnl was evaluated as a marker for: (1) the *Mucorales* family, (2) species identification (*Rhizopus arrhizus*, *R. microsporus*, *Mucor circinelloides*, and *Lichtheimia* species complexes), (3) growth stage, and (4) quantification. Sensitivity, specificity, discriminatory power, the limit of detection (LoD), and cross-reactivity were evaluated. Assays were tested using pure cultures, spiked clinical samples, murine organs, and human paraffin-embedded-tissue (FFPE) samples. Mitochondrial markers were found to be superior to nuclear markers for degraded samples. Rnl outperformed the UMD universal^®^ (Molyzm) marker in FFPE (71.5% positive samples versus 50%). Spiked blood samples highlighted the potential of rnl as a pan-*Mucorales* screening test. Fungal burden was reproducibly quantified in murine organs using standard curves. Identification of pure cultures gave a perfect (100%) correlation with the detected internal transcribed spacer (ITS) sequence. In conclusion, mitochondrial genes, such as rnl, provide an alternative to the nuclear 18 S rDNA genes and deserve further evaluation.

## 1. Introduction

The order *Mucorales* is assigned to the subphylum *Mucoromycotina*, one of the most ancient, early divergent groups of fungi [1]. In recent decades, the incidence of mucormycosis has increased due to: (a) a growing population of immunocompromised patients [2], (b) increased awareness [3], (c) the impact of mold-active prophylaxis [4], and (d) increasing numbers of patients with uncontrolled diabetes mellitus [5].

Until now, *Mucorales* have been identified using primarily histological and standard mycological methods, as well as DNA sequencing of the internal transcribed spacer (ITS) region [6]. These gold standard approaches are time-consuming and can be performed only for well-established infections. In contrast, molecular diagnostic tools can facilitate early diagnosis and screening. In medical mycology, a combination of real-time polymerase chain reactions (PCRs) and antigen-enzyme-linked immunosorbent assays (ELISAs) are used to supplement classical culture- and microscopy-based detection of pathogens in clinical specimens [7]. 

Discrimination of mucormycetes from other fungal agents is required for treatment decisions and patient management. Recent advances in the molecular diagnostics of mucormycetes have been comprehensively reviewed by Millon et al. [5], but key challenges such as standardization remain unmet [8]. The identification of pure cultures using matrix-assisted laser desorption ionization–time-of-flight mass spectrometry (MALDI-TOF MS) [9], plus the development of a novel pan-fungal antigen assay that detects a disaccharide in serum (MS-DS) and covers mucormycetes, are important developments [10]. Recent steps towards a pan-*Mucorales* marker system have included the use of real-time PCR with partial cytochrome B [11] as a template and a pan-*Mucorales* PCR that detects the spore coating protein (CotH) gene in urine samples [12,13]. Apart from these two examples that illustrate how pan-*Mucorales* detection can use alternative gene regions and markers, real-time PCR assays still use nuclear ribosomal DNA target (18 S, 28 S, and the internal transcribed spacer (ITS)) regions [14]. These ribosomal markers appear to be less useful for the design of pan-*Mucorales* markers because of a poor specificity at the order level. The potential of non-nuclear markers has yet to be to be evaluated.

Mitochondrial genes have long contributed to forensicscience [15]. Mitochondrial DNA is protected from degradation (possibly due to the mitochondria’s double membrane), is stable across multiple generations (due to absence of meiosis or genetic recombination events), and exists in higher copies numbers than nuclear DNA [16]. These properties have encouraged evaluation of mitochondrial genes as diagnostic markers in mucormycetes. 

Mucormycetes are hard to detect in clinical specimens (particularly biopsies) and their non-septate hyphae make quantification difficult in murine experiments [17]. Hyphae break easily during sample preparation and their leaked DNA is susceptible to degradation by host DNAses. In contrast, the high copy number and the degradation resistance of mtDNA [16] should confer increased marker sensitivity. Moreover, normalization of mitochondrial genes with a nuclear housekeeping gene should enable differentiation between resting and active fungal stages. This would be particularly useful for the interpretation of murine studies by enabling discrimination between colonization (i.e., presence of conidia) and infection (i.e., presence of hyphae) in respiratory tract samples. 

This study aimed to assess the potential of a mitochondrial gene marker (rnl) for a broad range of diagnostic and research applications. The rnl gene was tested as a marker for the following applications: (a) pan-*Mucorales* identification in paraffin-embedded tissues, (b) pan-*Mucorales* identification marker in whole blood samples, (c) *Mucorales* quantification in murine organs, (d) growth stage determination, and (e) identification of *Mucorales* in pure cultures. 

## 2. Materials and Methods

### 2.1. Strain Collection and Cultivation

The *Mucorales* strain sets used to evaluate the performance of the rnl marker given in Appendix A were characterized in a previous study [18]. Fungal stocks stored in 0.9% sodium chloride at −20 °C served as inoculum sources. Fungi were grown on supplemented minimal medium (SUP) agar [19] at 37 °C, with the exception of *Mucor* spp., which were grown at 30 °C and further incubated for 3–5 days. Mycelia were harvested in 0.9% sodium chloride. Conidia were separated using a 40 µm nylon cell strainer (VWR^®^, Belgium). Identification (ID) at the genus level was performed by visualization of microscopic (Olympus CX21 microscope; Olympus, USA) and macroscopic (Axioplan microscope; Zeiss, Germany) characteristics. Species ID was obtained by direct sequencing of the internal transcribed spacer (ITS) [20] with slightly modified primers [18]. Genomic DNA was extracted [21] and ITS sequences were identified using the pairwise sequence alignment tool of the central bureau schimmelcultures (CBS) Knaw database (http://www.cbs.knaw.nl/Collections). 

### 2.2. rnl Marker Design

The discriminatory power of the mitochondrial rnl gene was evaluated in silico using reference mitochondrial genomes of three *Mucorales* species: *Rhizopus arrhizus*, *Lichtheimia hongkongensis* (syn. *Lichtheimia ramosa*), and *Mucor circinelloides* f. *lusitanicus* (GenBank AY863212.1, KJ561171.1 and KR809877.1) (Figure 1). Universal primers for amplification and sequencing, rnl_fw (5′-GCGAAATACCTTGGCCACTA–3′), and rnl_rv (5′–CCGGCTTATGCCATTACACT–3′) were designed using Geneious™ software v. 8.1.9 (Biomatters Limited, Auckland, NZ) to give a DNA fragment amplified from positions 1925 bp to 2314 bp of the *R. arrhizus* rnl gene (GenBank AY863212.1). The PCR reaction mixture (10 µL) contained 300 nM of each primer, 5 µL of KAPA2 G Robust HotStart ReadyMix™ (KAPA Biosystems, Boston, Massachusetts, USA), 3 µL of ultrapure water, and 1 µL of genomic DNA (10–20 ng/µL). PCR amplification used a peqSTAR 2× gradient thermal cycler (PEQLAB biotechnology, Erlangen, Germany) programmed as follows: initial cycle 95.0 °C/2 min, followed by 30 cycles at 95.0 °C/15 s, 57.0 °C/30 s, 72.0 °C/1 min, and a final cycle at 72.0 °C/3 min. PCR amplicon sizes ranged from 367 bp (for *R. arrhizus*) to 391 bp (for *L. ramosa* and *M. circinelloides*). DNA sequence analysis of the PCR products used the BigDye^®^ Terminator v 3.1 Cycle Sequencing Kit (ThermoFisher Scientifics, Waltham, Massachusetts, USA) together with the BigDye XTerminator^®^ Purification Kit (ThermoFisher Scientifics, Waltham, Massachusetts, USA). The products were sequenced using an ABI 3730 XL automatic sequencer (ThermoFisher Scientifics, Waltham, Massachusetts, USA). Sequences were analyzed with Geneious™ software v 8.1.9 (Biomatters Limited, Auckland, NZ) to identify single nucleotide polymorphisms (SNPs). 

### 2.3. rnl Real-Time PCR (Pan-Mucorales Marker)

For rnl qPCR-high resolution melt (HRM) assay design, 82 rnl sequences representing *R. arrhizus*, *R. microsporus*, *M. circinelloides*, and *Lichtheimia* spp. complex (*L. corymbifera* and *L. ramosa*) were aligned with the *R. arrhizus* reference sequence (GenBank AY863212.1). Primers (2 rnl_127 fw: 5′–GGTGTAGAATACAAGGGAGTCGA–3′ and 2 rnl_250 rv: 5′–GGAGAAATCCGCCCCAGATAA–3′) generated an amplicon of 124 bp. These primers were tested in silico for cross-reactivity with human and not-targeted fungal DNA. The qPCR reaction contained 10 µL of SsoFast™ EvaGreen^®^ Supermix (Bio-Rad Laboratories Incorporated, Hercules, California, USA), 1 µL of 10 µM of each primer, 6 µL of ultrapure water, and 2 µL of DNA eluate. All samples were run in duplicate. As positive controls, a set of *n* = 4 previously identified representative *Mucorales* samples (*R. arrhizus*, *R. microsporus*, *M. circinelloides*, and *Lichtheimia* species complexes; AS119, F50, AS84, and FF18, respectively, Appendix A) were used as templates, with sterile water (B. Braun, Melsungen, Germany) serving as no template control (NTC). The qPCR reactions were conducted using a CFX96™ real-time PCR detection system (Bio-Rad Laboratories Incorporated, Hercules, California, USA) under the following conditions: 98.0 °C/2 min, followed by 40 cycles of 98.0 °C/5 s and 61.8 °C/10 s. 

### 2.4. rnl Real-Time PCR-HRM (Species-ID)

The species identification PCR reaction contained 10 µL of SsoFast™ EvaGreen^®^ Supermix (Bio-Rad Laboratories Incorporated, Hercules, California, USA), 1 µL of 10 µM of each primer (see rnl real-time PCR), 6 µL of ultrapure water, and 2 µL of each DNA sample (20.0 ng/reaction). All samples were run in duplicate with three biological replicates and with positive controls for each species or species complex (*R. arrhizus*, *R. microsporus*, *M. circinelloides*, and *Lichtheimia* species complexes; AS119, F50, AS84 and FF18, respectively, Appendix A). Sterile water (B. Braun, Germany) was used as NTC. The qPCR reactions used a CFX96™ real-time PCR detection system (Bio-Rad Laboratories Incorporated, Hercules, California, USA) under the following conditions: 98.0 °C/2 min, followed by 40 cycles of 98.0 °C/5 s and 61.8 °C/10 s. After denaturation at 98.0 °C/3 s and cooling at 70.0 °C/30 s, a melt curve was generated from 72.0 °C to 81.0 °C with 0.1 °C increments/5 s dwell time. Melt curves were processed and analyzed using Precision Melt Analysis™ software (Bio-Rad Laboratories Incorporated, Hercules, California, USA) to determine average melting temperatures (Tm) and to assign samples to predefined species-specific or species-complex-specific melt profiles.

### 2.5. DNA Extraction and rnl/tef qPCR (Quantitative PCR)

Dissected brains, lungs, livers, and kidneys from black six mice (C57 BL/6 J) were homogenized following an established in-house protocol (refer to submitted paper by Speth et al., PloS Pathogens). All animal experiments were approved by the National Committee for Animal Care of the Austrian Federal Ministry of Science, Research, and Economy (BMWFW) (approval number BMWFW-66.011/0110-WF/V3 b/2016). Organs in Whirl-Paks^®^ (VWR International BVBA, Belgium) frozen in liquid nitrogen were mechanically crushed and then homogenized in cooled 2 mL Safe Lock Eppendorf^®^ tubes (Eppendorf AG, Hamburg, Germany) using 0.5 mm steel grinding balls at 30 Hz for 30 s in a Mixermill 400 (Retsch, Haan, Germany). Five-point standard curves were prepared as follows: 20 mg of tissue from uninfected mice was spiked either with 20 μL 0.9% NaCl as a negative control or with 10^4^–10^8^
*Lichtheimia corymbifera* (CBS 109940) or *R. arrhizus* (CBS 126971) cells. DNA was extracted from tissue samples using the DNeasy^®^ Blood and Tissue Kit (Qiagen, Germany). The final elution volume of 200 μL contained a total of 15–30 ng of DNA. Two μL samples of the DNA eluate were used in the PCR reaction described above. The standard curve was established with 3 biological replicates and 3 technical replicates for each tissue type and fungus. The qPCR used the rnl real-time protocol above and R^2^ cut-off values were calculated. The fungal loads of samples were calculated using Bio-Rad CFX Manager 3.1 software with the 5-point standard curve as reference. Consistent results were obtained by analyzing 3 biological replicates for each sample. 

### 2.6. rnl-tef qPCR (Growth Stage Determination)

One strain each of *L. corymbifera* (AS41), *R. arrhizus* (KOG-D3), *M. circinelloides* (SPL-CR1), and *R. microsporus* (AS109) was tested. Samples of 1 × 10^6^ conidia/mL were incubated in 50 mL liquid SUP at 30 °C to generate germlings and early hyphae. In pilot experiments, time points for germlings and young hyphae were established based on strain growth kinetics (BioStation IM-Q, Nikon, Vienna, Austria). Fungal development was monitored microscopically (Stemi 2000-C, Carl Zeiss Microscopy, Germany) to verify the harvest time points for germlings and early mycelia. Each experiment used 3 biological replicates and 2 technical replicates. DNA from 1 × 10^6^ conidia, 1 × 10^6^ germlings, or 0.2 g mycelium was extracted as described by Möller et al. (1992) and spiked with background DNA extracted from human blood samples using the DNeasy^®^ Blood and Tissue Kit (QIAGEN GmbH, Düsseldorf, Germany), as described below for clinical sample mimics. Performance was evaluated for both active stages (germlings, hyphae) and a resting stage (conidia). Rnl real-time PCR was conducted using the following conditions: initial denaturation 95.0 °C/ 3 min, followed by 45 cycles of 95.0 °C/15 s, 58 °C/ 30 s, and an extension step at 70.0 °C/ 3 min. All samples were tested in technical duplicates. 

Tef (transcription elongation factor 1-nucleic marker) served as a housekeeping gene for the real-time PCR. Primers for amplification of the 169-bp-long amplicon were designed using Geneious™ software v. 8.1.9 (Biomatters Limited, Auckland, NZ) and tested in silico for specificity. TEF_468 fw: 5′–GGAGTTCGARACCCCCAAG–3′ and TEF_640 rv: 5′–CGGGTTTGACCRTCCTTGGA–3′ were applied in a separate qPCR for the samples under the same conditions as for the rnl marker, but with the annealing temperature changed to 61.0 °C [22]. The rnl and tef qPCR runs for the biological triplicates of spores, germlings, and hyphae were conducted in technical duplicates, as were the positive controls (for AS119, F50, AS84, and FF18) and negative controls using pure human background DNA. The fold change between rnl and tef cycle thresholds was calculated using the formula 2∆C_t_. Mean values and ranges of 2∆C_t_ were calculated for each growth stage. The 2∆C_t_ values were used to obtain correlation with the fungal growth stage.

### 2.7. rnl Real-Time PCR-HRM Evaluation of Pure Cultures

The specificity of the rnl real-time PCR-HRM assay was tested blind in comparison to ITS direct sequencing analysis for the 109 strains in our collection (Appendix A). All samples were normalized to a concentration of 20 ng DNA/reaction, and positive controls were included for each species or species complex level. The confidence intervals for auto-called clustering were determined using Bio-Rad Precision Melt Analysis™ software v 1.2 (Bio-Rad Laboratories Incorporated, Hercules, California, USA). Marker species specificity was tested in vitro with DNA extracts from other commonly occurring *Mucorales* species, namely *M. racemosus* strain (AS 42), *Phycomyces blakesleeanus* (CBS 284.35), *Cunninghamella bertholletiae* (CBS 187.84), *Syncephalastrum racemosum* (AS 29), and *Rhizomucor pusillus* (AS 45). Cross-reactivity with other non-*Mucorales* fungi was tested using DNA extracts from *Candida albicans* (472 ca), *C. tropicalis* (424 i), *C. glabrata* (218 i), *C. krusei* (124 i), *Aspergillus fumigatus* (AF293), and *A. terreus* (118). The limit of detection (LoD) was determined for *L. corymbifera* (AS10), *L. ramosa* (33.12), *R. arrhizus* (LLU-MA1), *R. microsporus* (75-10), *R. microsporus* (75-10), and *M. circinelloides* (60-10), as these are the species most commonly found in Europe [23]. Ten-fold serial dilutions (range: 5.0 ng/µL–50.0 fg/µL) of DNA samples were prepared in Endosafe^®^ LAL water (Charles River Laboratories, Inc., USA).

### 2.8. rnl Real-Time PCR-HRM Evaluation of Assay Performance in Samples Simulating the Human DNA Background of Clinical Specimens

The qPCR LoD of the ID tool was tested using clinical sample mimics based on DNA samples from 200 µL freshly drawn EDTA-treated human whole blood, extracted using the peqGOLD Blood DNA Mini Kit (Peqlab Biotechnologies GmbH, Germany), spiked with respective DNA samples of the five species under investigation; namely, *L. corymbifera* (AS41 A; AS41 B; AS10), *L. ramosa* (AS75 A; AS75 B; 02-10), *M. circinelloides* (SPL- CR1; MAL-D3; AS107), *R. arrhizus* (BOL-CR4; KOG-D3; CM5368), and *R. microsporus* (75-10; 41-10; 14-11). Ten-fold dilutions series was made with fungal DNA content of 0.001%–100% of total DNA (10 ng) and 0%–99.999% human DNA, respectively. The 100% human background DNA served as the negative control. To test for potential application in routine diagnostics, the extraction method used instead was the MolYsis™ Basic5 kit (Molzym GmbH and Co. KG, Bremen, Germany) and the rnl real-time PCR-HRM protocol was modified to a seminested real-time PCR-HRM (see below). 

### 2.9. rnl Real-Time PCR-HRM Evaluation of Assay Performance in Clinical Sample Mimics

For clinical sample mimics, aliquots of 200µL EDTA whole blood collected from healthy volunteer donors were inoculated with a ten-fold dilution series of 1 × 10^7^–1 × 10^2^
*R. arrhizus* (KOG-D3) conidia/sample. DNA extraction was performed for all samples, including non-spiked human blood as negative controls. The MolYsis™ Basic5 kit was used to recover the microbial lysate depleted of human DNA, followed by the DNeasy^®^ Blood and Tissue Kit (QIAGEN GmbH)) according to the manufacturer’s instructions. 

### 2.10. Seminested rnl Real-Time PCR-HRM for Species ID in Clinical Sample Mimics

DNA extracts were tested in the seminested real-time PCR-HRMA (high resolution melt assay), with the first PCR reaction using a master mix composition and parameters as described for rnl real-time PCR-HRM. The amplicons recovered were diluted 1:300 with sterile water (B. Braun, Germany) and amplified using the same master mix composition in a real-time PCR run according to the following protocol: initial denaturation at 98.0 °C/1 min, followed by 20 cycles of 98.0 °C/5 s and 61.8 °C/30 s; denaturation at 98.0 °C/30 s and cooling at 70.0 °C/30 s; and generation of melt curve from 72.0 °C to 81.0 °C, with 0.1 °C increments/5 s dwell time. Subsequent analyses were performed as described for rnl real-time PCR-HRM. Species ID control DNAs for the four *Mucorales* species were extracted using the same procedure and added to each run. All experiments used biological duplicates and technical duplicates.

### 2.11. Paraffin-Embedded Tissue (FFPE) of Patients with Proven Fungal Infection and Suspected Mucormycosis

Samples of biopsies and autopsies were provided by the biobank of the Department of Pathology at the Innsbruck Medical University. The specimens were procured with the approval of the ethics committee (EK 122/04) from patients who gave informed consent. Specimens had been fixed in formalin and embedded in paraffin. Fixed blocks were cut into 3.0-µm-thick tissue sections and histologically validated by staining with Grocott and hematoxylin-eosin (HE). Slides were digitized using a Panoramic SCAN digital slide scanner (3 DHISTECH, Budapest, Hungary) with a Plan-Apochromatic objective (magnification: 20×, numerical aperture: 0.8). The histological evaluation and the scoring of the fungal infection were done using Panoramic Viewer software (3 DHISTECH, Budapest, Hungary). All tissues were reviewed and screened for mucormycosis. Grocott and hematoxylin-eosin (HE) staining detected hyphae typical of mucormycetes. These hyphae were non-septate or sparsely septate and showed an irregular, ribbon-like appearance. The angle of branching was variable and included wide-angle (90°) bifurcations. The following inclusion criteria for the study were used: (1) diagnosis of invasive mycoses based on histological and clinical findings, (2) diagnosis of mucormycosis based on histological features, and (3) confirmation of the causative agent of mucormycosis by real-time polymerase chain reaction (PCR) or positive culture.

DNA from 3 × 10 µm formalin-fixed, paraffin-embedded (FFPE) tissue sections per sample was extracted using the BioRobot EZ1 (Qiagen) and the EZ1 DNA Tissue Kit (Qiagen), according to manufacturer’s instructions (protocol: Purification of DNA from Paraffin-Embedded Tissue). The elution volume was adjusted to 50 µL. PCR positive samples were purified using the QIAquick PCR purification kit (Qiagen), according to the manufacturer’s instructions. The elution volume was adjusted to 30 µL using ultrapure water. Eluates were diluted 1:100 in Endosafe^®^ LAL water (Charles River Laboratories, Inc., USA). Twelve µL of the master mix, consisting of 10 µL SsoFast™ EvaGreen^®^ Supermix (Bio-Rad Laboratories Incorporated, Hercules, California, US) and 1 µL of 500 nM of each rnl primer (rnl real-time PCR), was added to 8 µL of the DNA eluates. Rnl real-time PCR was run in duplicate.

## 3. Results

### 3.1. rnl Marker

The mtDNAs of the targeted mucormycetes species (*Rhizopus arrhizus*, *R. microsporus*, *M. circinelloides*, and *Lichtheimia* species complexes) are between 54,178 bp for *R. arrhizus* [24] and 31,830 bp for *L. ramosa* [25], including the 2878-bp-long rnl gene that encodes the large subunit ribosomal RNA. The gene was selected for *Mucorales* species identification because of its potential discriminatory power due to 15 single nucleotide polymorphisms (SNPs) found within a specific 279 bp fragment (Figure 1 and Appendix A) used as the molecular marker.

### 3.2. Marker Performance in Pure Cultures

The 279 bp partial fragment of the rnl gene was tested as a pan-*Mucorales* marker and species ID marker for the most frequently occurring European mucormycetes, namely R. arrhizus, *R. microsporus*, *M. circinelloides*, *L. corymbifera*, and *L. ramosa*. The temperature of 61.8 °C was optimal for annealing, with an amplification efficacy of 100% for all species tested (Appendix A). The table shows that the sibling species *L. corymbifera* and *L. ramosa* had identical SNP profiles. They were, therefore, referred to as the *Lichtheimia* spp. complex (Figure 2). 

To optimize the ID tool, a total of 46 isolates (*R. arrhizus* (*n* = 10), *R. microsporus* (*n* = 10), and *M. circinelloides* (*n* = 10); *L. corymbifera* species complex (*n* = 16)) were tested. The values obtained for the average melt temperature (Tm) ± SD were evaluated in ten experiments (Figure 3). The lowest average Tm was observed for *R. microsporus* (77.78 ± 0.19 °C), followed by *R. arrhizus* (78.18 ± 0.11 °C), the *Lichtheimia* spp. complex (78.55 ± 0.12 °C), and *M. circinelloides* (79.11 ± 0.11 °C) (Figure 3C). Both the normalized melt curves (Figure 3A) and difference curves (Figure 3B) distinguished the four taxonomic entities correctly at the species or species complex level, respectively. Blind-tested, unknown samples (two per species) were clustered correctly with the melt profile of their respective positive controls, enabling identification. To comprehensively assess the accuracy of identification using the rnl real-time PCR-HRM, a comparative analysis was conducted with ITS sequencing using a total of 109 (*L. corymbifera* (*n* = 31), *L. ramosa* (*n* = 13), *R. arrhizus* (*n* = 23), *R. microsporus* (*n* = 25), and *M. circinelloides* (*n* = 17)) strains (Appendix A). Species-dependent Tm and confidence ranges for reliable assignments were determined (Figure 3C). The average Tm for each species was distinct (Appendix A). The lowest values were found for *R. microsporus* (77.79 ± 0.09), followed by *R. arrhizus* (78.10 ± 0.11), *Lichtheimia* spp. complex (*L. corymbifera* 78.5 ± 0.07 and *L. ramosa* 78.55 ± 0.09), and *M. circinelloides* (79.09 ± 0.08). Confidence values reaching 100% were achieved for all species (*L. corymbifera* (96.90%–100%), *L. ramosa* (99.10%–100%), *R. microsporus* (98.10%–100%), *M. circinelloides* (97.60%–100%)), except *R. arrhizus*, whose highest value was 99.90% (97.10%–99.90%) (Appendix A). 

Analysis of the rnl real-time PCR-HRM confirmed pan-*Mucorales* species specificity. Preliminary in silico screens identified overlaps with 11 other mucoralean sequences deposited in the NCBI database, confirming rnl conservation within the *Mucorales* order. In vitro testing found that a wide variety of mucormycetes could be detected using rnl real-time PCR-HRM. Among the species tested were: *C. bertholletiae*, *M. racemosus*, *P. blakesleeanus*, *R. pusillus*, and *S. racemosum*. All species gave distinctly different melt peaks and melt profiles. Unspecific clustering among the four target species was not observed (Appendix A). 

Cross-reactivity with non-mucoralean fungal species and the human and murine genomes was excluded in silico. Cross-reactivity in routine diagnostics with commonly encountered ascomycetes, namely *Candida* and *Aspergillus* species, was excluded using additional in vitro tests. Cross-reactivity with human and murine DNA was also excluded by testing those DNAs in vitro. Rnl real-time PCR-HRM did not amplify *C. albicans*, *C. glabrata*, *C. krusei*, *C. tropicalis*, *A. fumigatus*, and *A. terreus* samples (i.e., no cross-reactivity of the rnl marker was observed) (Appendix A). Thus, the rnl marker is a highly specific pan-*Mucorales* marker. The limit of detection (LoD) for rnl real-time PCR was found to be 50.0 fg/µL per reaction, equating to one genome copy (C_t_ value of 33.17 ± 1.64 for one *R. arrhizus* genome, 37.54 ± 0.32 for one *R. microsporus* genome, 34.65 ± 0.68 for one *M. circinelloides* genome, and 36.91 ± 1.16 for one *Lichtheimia* spp. genome) (Figure 4A,B). For species identification via rnl real-time PCR-HRM analysis, 400–500 fg DNA (equal to ~10 genomes; genome size of *R. arrhizus* is 45.26 Mbp, *R. microsporus* is 25.97 Mbp, *M. circinelloides* is 36.6 Mbp, and *Lichtheimia* spp. complex are 33.6 Mbp) was needed for identification with a confidence of 90.50%–100%. C_t_ values for 10 genome copies were 32.49 for *R. arrhizus*, 35.25 for *R. microsporus*, 28.97 for *M. circinelloides*, and 32.70 for *Lichtheimia* spp. complex (Figure 4C,D). 

### 3.3. Rnl Marker Evaluation with Human DNA Background

Clinical specimen mimics consisted of human background DNA extracted from EDTA whole blood spiked with fungal DNA. The LoD for rnl real-time PCR detection was found to be 100.0 fg mucoralean DNA/initial sample for all species of interest, with cycle thresholds displaying average values of 36.35 ± 0.60 for *L. corymbifera*, 36.69 ± 0.46 for *L. ramosa*, 34.76 ± 0.92 for *R. arrhizus*, 35.24 ± 0.43 for *R. microsporus*, and 33.47 ± 0.63 for *M. circinelloides*. The ranges of linearity differed between the species. The broadest range of linearity was seen with *M. circinelloides* (0.1 pg–1.0 ng fungal DNA/µL total DNA) and *L. corymbifera* (1.0 pg–10 ng fungal DNA/ µL total DNA), followed by *R. arrhizus* (1.0 pg–1.0 ng fungal DNA/ µL total DNA), *L. ramosa* (100.0 pg–1.0 ng fungal DNA/ µL total DNA), and *R. microsporus* (100.0 pg–1.0 ng fungal DNA/ µL total DNA). The coefficients of determination were all greater than 0.99 (Appendix A). 

### 3.4. Seminested Real-Time PCR-HRM for Conidia-Spiked, EDTA-Treated Human Blood Samples

When the rnl real-time PCR-HRM assay was tested using conidia-spiked human blood samples, positive amplification to a concentration of 1 conidia/200 µL blood sample (corresponds to 0.1 conidia/PCR reaction; 2 µL of DNA eluate were used from the total 20 µL DNA eluate) was achieved, with an average C_t_ value of 36.98. Several melt curves of the serial dilution were linear at this concentration (Appendix A). Identification based on the HRM profile was possible down to 10^4^
*R. arrhizus*-conidia-spiked in 200 µL blood, equating 10^3^
*R. arrhizus* conidia/PCR reaction (Figure 5). Lower concentrations of fungal DNA could not be assigned to the correct melt profile. 

### 3.5. rnl Real-Time qPCR for the Detection of Fungal Burden in Murine Organs

A murine organ model was established to provide standardized samples of kidney, spleen, liver, lung, and brain tissue for determination of fungal burden/20 mg tissue using rnl qPCR. Data are given in Figure 6. We obtained linear trend lines for the ten-fold dilution series of 10^8^ to 10^4^ spores per sample. *L. corymbifera*-spiked samples led to coefficients of determination R^2^ > 0.95 for brain and lung, R^2^ > 0.90 for kidney, and R^2^ > 0.85 for liver and spleen; whereas *R. arrhizus* spiked samples gave R^2^ > 0.95 for kidney, brain, and lung, and R^2^ > 0.90 for liver and spleen. The consistent results delivered by rnl qPCR show that rnl is a robust marker for fungal burden evaluation (Appendix A). Independent evaluation by other laboratories should now be a priority.

### 3.6. Growth Stage Determination Using rnl/tef qPCR

The ability of the rnl marker to identify resting and active growth stages of *Mucorales* in clinical sample mimics was investigated. The activity stage was detected using the fold change (2∆C_t_) between the rnl mitochondrial marker and the nucleic tef housekeeping gene. Mitochondrial DNA occurs in larger amounts in active fungal cells than in resting conidia. The rnl/tef qPCR found that C_t_ values obtained by rnl primers were lower than the ones obtained by tef primers. The sensitivity of rnl marker was higher than the tef marker, with an average cycle difference of 2.26 for conidia, 3.32 for germlings, and 3.20 for hyphae. This confirms higher abundance of mtDNA compared to nDNA present in the samples (Appendix A). Fold changes of rnl compared to tef were found to be 4.98 in spores (minimum, 3.99; maximum, 5.52), 10.75 in germlings (minimum, 8.77; maximum, 12.82), and 8.61 in hyphae (minimum, 6.95; maximum, 11.06). These results define a cut-off value of 5.60 that differentiates between the resting (spores) and active growth (germlings, hyphae) stages of mucormycetes (Figure 7).

### 3.7. rnl as a Pan-Mucorales Marker in FFPE Tissue Samples

The performance of rnl as a pan-*Mucorales* marker was evaluated for FFPE tissue samples. The rnl real-time PCR assay was carried out on 21 clinical samples previously diagnosed with suspected *Mucorales* infections via culture and microscopic examination. Of 21 samples, 15 (71.4%) were pan-*Mucorales* positive compared to 8 (50%) of 16 samples that tested positive by UMD Universal^®^ (Molzym) pan-fungal PCR (commercial kit used in routine diagnostics). Four samples found positive with rnl real-time PCR were negative with UMD Universal kit (Molzym). These included two autopsy samples with insufficient quality DNA for the UMD Universal kit. One sample not detected by pan-*Mucorales* PCR was found to be positive by the pan-fungal PCR, and Candida spp. was detected as the causative species. Another sample that was negative with the pan-*Mucorales* marker was not tested with the UMD Universal kit. Only its histology was indicative of a fungal infection but that might have been caused by an ascomycete or basidiomycete. Detailed results are given in Table 1. 

## 4. Discussion

Patients suffering from mucormycosis exhibit signs and symptoms similar to patients suffering from other invasive fungal infections, such as aspergillosis [26,27]. The ability to diagnose mucormycosis separately from other fungal infections is essential for therapeutic management and to improve patient outcomes. In contrast to mucormycosis, the primary therapy for aspergillosis and scedosporiosis is voriconazole, a short-tailed azole drug that is ineffective against mucormycetes [28,29]. In contrast, the primary therapy for mucormycosis is amphotericin B [30]. Therefore, diagnostic tools that distinguish mucormycosis from other fungal infections are needed urgently [31].

Current diagnostics for mucormycetes primarily use direct imaging, histopathology, culture, and direct examination [32]. These follow the recommendations of the International Society for Human and Animal Mycology working group, which has also suggested ITS sequencing as the gold standard [33]. Despite the development of several PCR-based assays, these have focused on different genera or species [17] and have almost exclusively involved nuclear ribosomal DNA targets (18 S, 28 S, and internal transcribed spacer (ITS)) [34,35,36,37,38,39,40]. However, nuclear ribosomal genes appear to be inadequate targets as pan-*Mucorales* markers. The ITS region exhibits intraspecific variability within 3.24% of the *Mucoromycotina* [41]. Moreover, heterogeneity of ITS 1 and ITS 2 sequences has been detected in *R. microsporus* [42]. Additionally, the discriminatory power for other fungal pathogens at the order level is too limited. We rationalized that a new approach that discriminates mucormycosis from aspergillosis and candidiasis could enable prompt diagnosis and customized antifungal treatment. So far, only two pan-*Mucorales* markers that identify clinical specimens have been published. The first is a PCR technique targeting the spore-coating encoding protein (CotH) gene [12,13], and is mainly used for urine samples. The commercial PCR MucorGenius^®^ was launched recently by PathoNostics^®^, but for research purposes only. Unfortunately, the molecular basis of the test is not available (https://www.pathonostics.com/product/mucorgenius) and the assay is yet to be validated on clinical specimens. Some authors have also suggested a combinational approach, using different biomarkers for early diagnosis of mucormycosis [8].

Pan-*Mucorales* detection may require alternative gene regions that are conserved within the mucormycetes and distinctly different from ascomycetes and basidiomycetes. This report has investigated rnl (Figure 1) as an example of a mitochondrial marker. We first evaluated its suitability as a marker for: (1) pan-*Mucorales* identification, (2) species identification (*Rhizopus arrhizus, R. microsporus, Mucor circinelloides,* and *Lichtheimia* species complex), (3) growth stage determination, and (4) quantification. The mitochondrial gene has the following advantages for the detection and identification of *Mucorales*: (1) high copy number of mitochondrial genes per fungal cell; (2) enhanced stability due to the mitochondrial double membrane, increasing the chance it can be extracted despite low quality or highly degraded nuclear DNA in samples [43,44]; (3) lower sequence heterogeneity than ITS sequences [42], and as our results show, no intraspecific variability. The rnl gene was found to be a suitable candidate marker gene because a universal primer pair could be designed to select a well-conserved region within the gene, despite the presence of introns in fungal mitochondrial genomes [43]. The primer pair enabled the establishment of all the rnl-based PCR formats reported herein (see Materials and Methods). The primer set provides a common core reaction for PCR formats for in silico and in vitro studies, and analysis of clinical specimen mimics, animal tissues, and FFPE. The primer set was evaluated in silico and in vitro for its cross-reactivity with human and murine DNA. No cross-reactivity was detected, even at very high DNA concentrations (10 ng/PCR reaction). 

In a second phase of testing, cross-reactivity with other non-*Mucorales* fungal pathogens was evaluated in silico because precise and clear discrimination of *Mucorales* from human pathogenic ascomycetes (e.g., *Aspergillus* and *Candida)* and basidiomycetes (e.g., *Cryptococcus*) is essential for targeted treatment [17]. As *Aspergilli* and *Candida* spp. are the most common agents of fungal infections, representative species were also tested also in vitro. Primer dimers were found above C_t_ > 38 for *C. glabrata*, *A. fumigatus* (Appendix A), and in the no template control (NTC). A cutoff value for positivity calling of C_t_ 38 is recommended for rnl real-time PCR. Hence, cross-reactivity for *A. fumigatus* and *A. terreus*, as well as *C. krusei, C. tropicalis, C. glabrata,* and *C. albicans*, could be excluded. The amplification of rnl from other mucormycetes was confirmed in silico and is shown in Appendix A. Key measures, such as LoD of the assay for pan-*Mucorales* positivity and species identification, were one genome copy and ten genome copies, respectively (Figure 4). 

The potential of rnl as a marker suitable for species identification was confirmed for four of the most common causes of mucormycosis in Europe (*R. arrhizus, R. microsporus, M. circinelloides*, and *Lichtheimia* species complex) (Figure 2). The two sibling species, *Lichtheimia corymbifera* and *L. ramosa*, could not be distinguished. This is not a clinical limitation, as there is no evidence of differences in resistance or virulence [45]. The basis for HRM identification is 15 SNPs present in the partial rnl fragment (Appendix A). The rnl real-time PCR-HRM delivered the same results as ITS sequencing (Figure 2 and Appendix A). Real-time PCR-HRM assays for species identification have previously been established for bacteria [46,47], protozoans [48], and other fungi [38,49,50,51,52,53,54,55]. Bialek et al. (2005) [56] published an alternative PCR-HRM approach using 18 S rDNA primers and an intercalating dye, similar to our study. We identified the most common European mucormycetes in pure cultures and clinical specimens. The identification of pure cultures is, therefore, feasible and should be further developed (Figure 3 and Appendix A). For pure cultures, the method is ideal for rapid in-house testing. It does not require databases or expensive equipment, has high sensitivity due to the real-time format using intercalating dyes, and has low setup costs. For species identification, the LoD for assays was approximately 10 genome copies/mL in spiked whole blood samples (Figure 5). Even though the LoD for species identification was 10-fold higher than for the pan-*Mucorales* marker only, previous studies suggest the fungal DNA load in serum samples of patients suffering from invasive mucormycosis is approximately 10- to 100-fold higher than in patients suffering from aspergillosis [57]. This should facilitate species identification in patient blood samples. The testing of blood samples from patients with invasive mucormycosis is now needed to assess the performance of the assay in routine diagnostic settings. All published real-time PCR methods that detect the causative agent on a genus level [39,57,58] or species level require subsequent sequencing of the PCR product [59]. Blood-based screening PCRs are especially useful for high-risk patients. The PCR-HRM assay is expected to diagnose the most common causative agents within one working day in pure cultures and blood samples, without the need for DNA sequencing. 

The rnl marker was evaluated as a pan-*Mucorales* marker in FFPE samples (Table 1). Fifteen out of 21 samples (71.4%)) were positive with the pan-*Mucorales* marker. However, the previous diagnosis *Mucorales*-positive infection in two of the negatively tested samples appears questionable. One was found *Candida*-positive by the pan-fungal PCR (suggesting co-infection or infection by *Candida* alone). The other sample was described by histology only as an invasive fungal infection, without further evidence of mucormycosis. Exclusion of these two samples makes the sensitivity 78.9%. A limitation of our study was that fresh tissue samples were not available. It is known that, independent of the assay format, analytic sensitivity of fresh tissue is between 97%-100%, and for FFPE only is between 56% and 80% [5]. Our rnl real-time PCR assay needs to be evaluated using fresh tissue samples to assess its full potential and to determine its specificity and sensitivity. Application of the two different PCRs (UMD universal Molzym and rnl real-time PCR) to the same clinical material showed that the sensitivity of our method at 71.4%–78.9% outperformed the 50% sensitivity of the commercial assay. The mitochondrial markers may be especially useful for necrotic tissues because of their likely resistance to degradation. In particular, the rnl-based PCR detected rnl in two out of three samples and outperformed the 28 S rDNA based Molzym assay, which failed to detect its target in the same autopsy samples. Studies on comprehensive sample sets are needed to confirm these preliminary results. 

The rnl-qPCR successfully quantified fungal burden in murine organs (brain, lung, liver, and kidney) using species-specific standardization curves generated for *Lichtheimia corymbifera* and *Rhizopus arrhizus* (Figure 6). Its high R^2^ (85-95%) makes rnl a robust marker for the determination of fungal burden. For pulmonary infection models, rnl/tef qPCR can be used to determine both fungal burden and fungal growth stage. This should allow, especially in pulmonary infection models, differentiation between resting conidia and active proliferating hyphae. To the authors’ knowledge, rnl provides the first growth stage marker for a fungus based on real-time PCR assay. The growth stage determination was possible in both pure cultures and clinical sample mimics (Figure 7). The assay uses the principle that mitochondrial biogenesis differs between the resting and active cell stages and results in a variation in the ratio of mitochondrial DNA molecules and nuclei [60]. Germlings and hyphae require more energy than spores as a resting stage. The greater mitochondrial abundance found in active stages is paralleled by an increased number of rnl genes per cell (Figure 7). This elevation in template levels lowers cycle thresholds for the correspondent samples in qPCR. The chosen internal control gene, tef, is located in the nuclear DNA, and is, therefore, proportional to the presence of nuclei. The changing relationship between nuclei and mitochondria enables discrimination between the different growth stages. Resting stages had a 2∆C_t_ < 5.60 and active stages had a 2∆C_t_ > 5.60. In our experiments, this value discriminates between the growth stages for all mucormycetes tested, namely *R. arrhizus, R. microsporus, M. circinelloides,* and *Lichtheimia* species. This cut off value might vary between PCR platforms, experimental conditions, and fungal species tested. As this is the first study to use this approach, we look forward to future comparative data from other laboratories. The ability to discriminate between the various growth stages may be of interest for other human pathogenic fungi; for example, aspergilli in respiratory tract samples such as bronchoalveolar lavages, where discrimination between colonization and infection during aspergillosis remains a significant challenge [61]. 

## 5. Conclusions

Our data show rnl, a representative gene of mtDNA, is a promising marker that can be used for various research and diagnostic applications. The rnl fragment marker identifies the most common mucormycete species (*R. arrhizus, R. microsporus, M. circinelloides,* and *Lichtheimia* spp. complex) in pure cultures. For clinical specimens, the marker’s main strength is its ability to serve as a pan-*Mucorales* marker. For tissue samples and highly degraded samples (autopsy samples), mitochondrial gene-based assays appear to be a promising, more robust alternative to the classical 18 S rDNA- and 28 S rDNA-based assays. The rnl marker provides consistent quantification of fungal burden in murine organs. Differing from nuclear markers, mitochondrial markers have the unique capacity to indicate growth stage, which could also be demonstrated in clinical specimen mimics. Other non-nuclear markers deserve consideration because they could improve the capacity to effectively diagnose fungal infections at the species level. 

## Figures and Tables

**Figure 1 jof-05-00098-f001:**
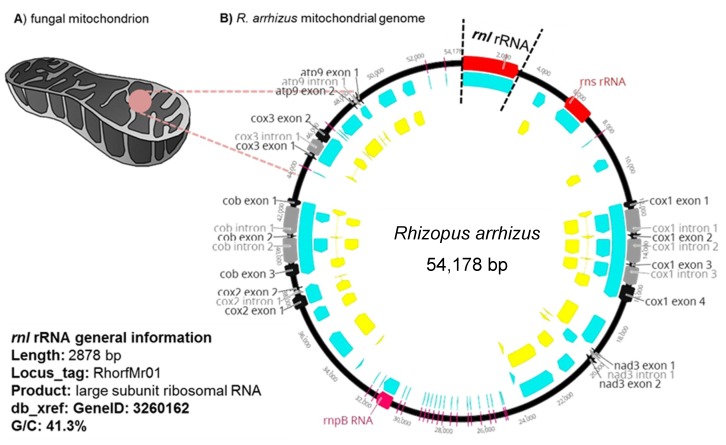
Location of the rnl gene (a large subunit of the ribosomal RNA) in the mitochondrial genome of *R. arrhizus.* (**A**) shows a fungal mitochondrion. (**B**) provides an overview on the mitochondrial genome.

**Figure 2 jof-05-00098-f002:**
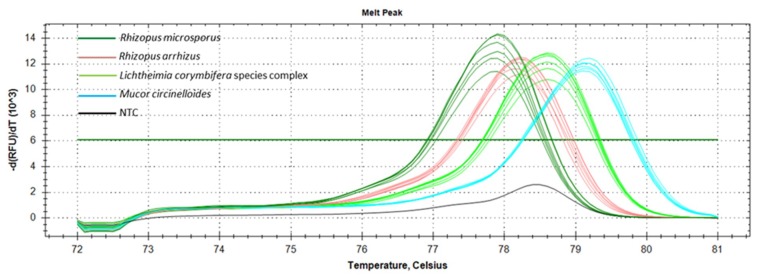
Overview of the melt peaks obtained by rnl real-time PCR-HRM for the targeted *Mucorales* species and species complexes (*Rhizopus arrhizus*, *R. microsporus*, *Mucor circinelloides*, and *Lichtheimia* species complexes). Note: NTC = no template control.

**Figure 3 jof-05-00098-f003:**
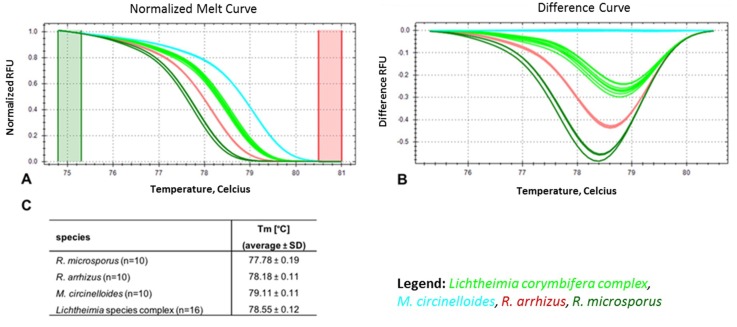
Melt curves: (**A**) high resolution melt curve, (**B**) melt difference curve (Tm) with respective standard deviation (SD), and (**C**) melt temperatures obtained for DNA eluates of *R. microsporus*, *R. arrhizus*, *M. circinelloides*, and *Lichtheimia* spp. pure cultures using rnl real time PCR-HRM.

**Figure 4 jof-05-00098-f004:**
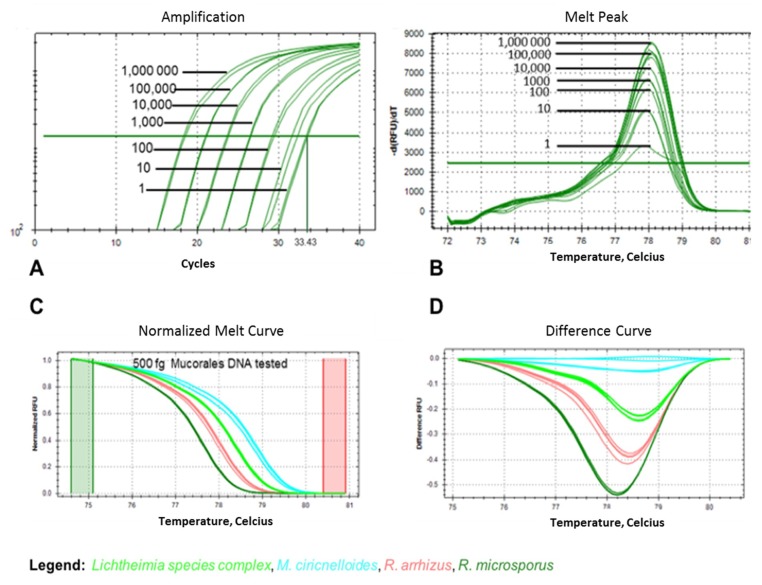
Limit of detection of the rnl real-time PCR-HRM in pure cultures. (**A**) Amplification of the rnl fragment for each genome copy number. (**B**) Melt peaks for each genome copy number. (**C**) HRMA melt curves for 500 fg of tested mucoralean DNA from all species under investigation. (**D**) Difference curve for species identification.

**Figure 5 jof-05-00098-f005:**
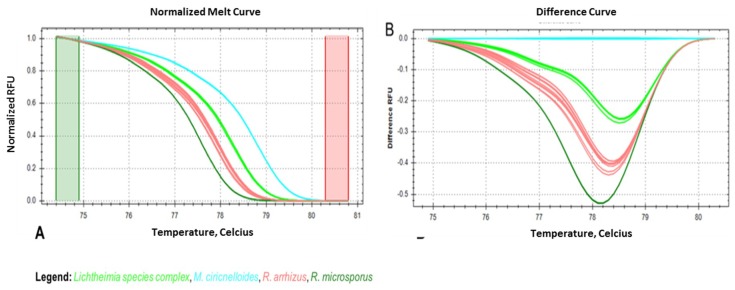
Sensitivity of the rnl real-time PCR-HRM assay for human EDTA-whole blood samples spiked with *R. arrhizus* conidia in serial dilutions. (**A**) Normalized melt curve and (**B**) difference curves for individual species.

**Figure 6 jof-05-00098-f006:**
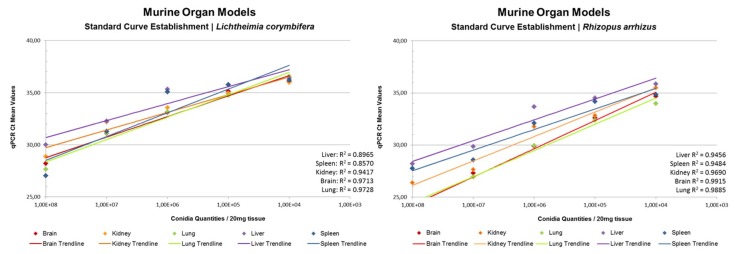
Standard curve obtained for murine organ models of conidia-spiked brain, kidney, lung, liver, and spleen. Values are given per 20 mg tissue. (**A**) Standard curves for *L. corymbifera*. (**B**) Standard curves for *R. arrhizus*. Standard deviations are given in Appendix A.

**Figure 7 jof-05-00098-f007:**
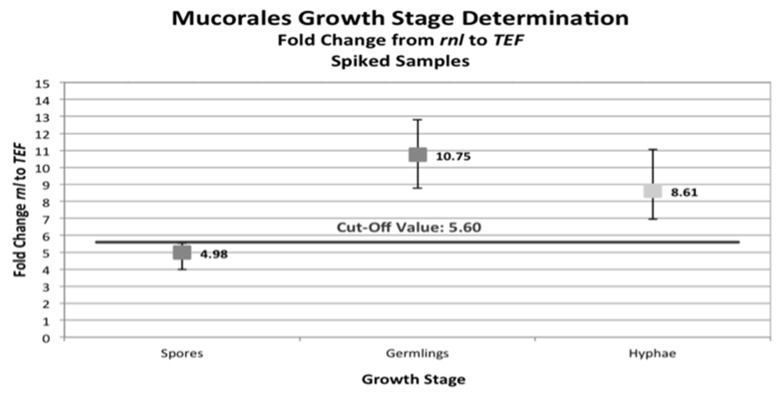
Growth stage determination of the *Mucorales* species investigated. Minima, medians, and maxima of the fold-change evaluation between the mitochondrial rnl marker and nuclear tef marker are plotted against the corresponding growth stages. The defined cut-off value of 5.60 ascertained via 2ΔC_t_ differentiates the resting (spores) and active (germlings or hyphae) developmental stages.

**Table 1 jof-05-00098-t001:** Results from paraffin-embedded tissue (FFPE) samples from patients with proven fungal infection and suspected mucormycosis.

			Histology		Routine Diagnostic PCR		Novel PCR
Patient ID	Sample Type	Tissue Type	Histology/Pathology Diagnosis	Microscopy (Grocott Staining)	Pan-Fungal PCR	Culture or Pan-Fungal PCR ID	Pan-*Mucorales*
TR	histology	lung	necrotizing mucormycosis with tissue invasion	(+); mucormycosis	(-)	no ID	(-)
HJ	histology	lung	inflammatory reaction with macrophages of connective tissue, with embedded fungal structures, most probably mucormycetes	(+); mucormycosis	(-)	no ID	(-)
KA	histology	lymph node, lung	necrotizing pneumonia caused by a fungus	(-); single unidentifiable hyphae	(-)	no ID	(-)
VM	histology	skin ulcer	abscess building panniculitis due to sepsis with fungi, suspected mucormycosis	(+); mucormycosis	not performed	not available	(+)
PG	histology	lung	lung tissue with bronchiolitis obliterans with organizing pneumonia (BOOP) and fungal infection (mucormycosis)	(+); fungal elements suspected mucormycosis	not performed	*Rhizopus arrhizus* (culture)	(+)
WC	histology	jaw	necrotic, inflammatory connective tissue with evidence of fungi, correlating to a mucormycosis	(+); mucormycosis	(+)	*Candida* sp., *Cryptococcus* sp.	(+)
HI	histology	maxillary sinus	extensive fungal infection of sinus maxillaris	Grocott stain not performed	not performed	not available	(+)
ML	histology	lung	mucormycosis of lingula	(+); mucormycosis	(-)	mucormycete (culture)	(+)
LR	histology	lung	fungal structures	Grocott stain not performed	not performed	not available	(-)
EJ	histology	skin ulcer	suspected mucormycosis	(+); mucormycosis	(+)	*Candida* spp.	(-)
NN	histology	stomach	the material of ulcer and fungal structures, invasive mucormycosis	(+); no septate hyphae, mucormycosis	(+)	*Rhizomucor pusillus*	(+)
BH	histology	the soft tissue of groin	both samples fat tissue necrosis of the soft tissue	Grocott stain not performed	(+)	*Rhizopus microsporus*	(+)
BH	histology						(+)
PN	autopsy	pleura aspirate	invasive mucormycosis with organizing pneumonia; epicardial myocarditis with fungal elements; two “Kissing” ulcers in the stomach with a fungal infection, suspected mucormycosis	(+); no septate hyphae; mucormycosis	(+)	*Rhizopus arrhizus*	(+)
GS	autopsy	lung	irreversible lung collapse due to mucormycosis	Grocott stain not performed	(+)	*Lichtheimia* spp.	(+)
WE	autopsy	lung	invasive fungal infection, suspected aspergillosis or mucormycosis	(+); no septate hyphae, mucormycosis	(+)	*Rhizomucor pusillus*	(+)
TM	autopsy	lung	fungal pneumonia, suspected aspergillosis	Grocott stain not performed	insufficient DNA quality	not available	(+)
BB	autopsy	lung	invasive sepsis due to generalized invasive fungal infection	(+); no septate hyphae, mucormycosis	insufficient DNA quality	*Rhizopus microsporus (culture)*	(-)
LRo	autopsy	lung	multiple infected sides, suspected mucormycosis	(+); no septate hyphae, mucormycosis	insufficient DNA quality	not available	(+)
SI	autopsy	lung	periphery pulmonary embolism and pulmonary infraction including fungal elements, suspected mucormycosis	(+); no septate hyphae, mucormycosis	(-)	*Lichtheimia corymbifera*	(+)
VL	autopsy	tissue of the cheek	fungal burden, mucormycosis, evidence of mixed yeast and mucormycete infection within pulmonary tissue	(+); no septate hyphae, mucormycosis	(+)	*Lichtheimia corymbifera*	(+)

Insufficient DNA quality: internal control failed; Grocott stain not performed: only hematoxylin-eosin (HE) stain was performed; pan-fungal PCR: Molzym UMD universal PCR; Culture and/or pan-Fungal PCR ID: 18 S rRNA sequencing approach of positive PCR reactions or ITS sequencing of positive cultures; not performed: e.g.,, not enough material, culture was already positive; culture ID: sequencing of the ITS region; pan-Mucorales PCR: in-house established method see M&M; no ID: pan- fungal PCR was negative and therefore no ID is possible; pan-fungal PCR: Molzym UMD Universal.

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
