# Peer review of "Evaluation of a Novel Mitochondrial Pan-Mucorales Marker for the Detection, Identification, Quantification, and Growth Stage Determination of Mucormycetes"

_jof, 2019, doi:10.3390/jof5040098_

Round 1
Reviewer 1 Report
The difficult and often delayed diagnosis remains a major obstacles to reduce the mortality of mucormycosis. There is lack of biomarkers efficiently detecting Mucorales from a broad spectrum of clinical matrices and many of the existing markers lack the ability to reliably distinguish Mucorales from other pathogenic molds. Therefore, Caramalho and colleagues evaluated the potential of a novel mitochondrial molecular marker (rnl-qPCR assay). The study has been very thoroughly designed and executed and conclusions are supported by an impressive amount of validation experiments and supplementary data. The ability to distinguish resting and germinated growth stages as part of a molecular assay represents a significant advancement. Overall, the assay is of major interest for both translational/clinical mycology and pre-clinical studies in animal models. My only major concern refers to the relatively high limit of linearity in murine organ models (see comments below).
However, the manuscript would greatly benefit from careful revisions by a native speaker, as there is a multitude of grammatical errors and poorly constructed sentences (esp. in the Discussion section). Similarly, some of the figures are not very well designed and, at times, labels are difficult to read (e.g. in Figure 6). More consistent formatting, larger fonts, and a better resolution wound enhance the presentation of the figures.
Specific comments:
Multiple figures are numbered wrongly. Line 302: Table S1 does not provide any information regarding comparative outcomes of ITS and rnl assays. Lines: 345-348: Please carefully double-check both the text and figure. The diagrams do not seem to fully correspond with the ranking and numbers provided in the test. In panel S3B, the limit of linearity remains unclear, since the end of the regression line and shaded area do not match. In addition, the numbers are difficult to appreciate, as the text refers to fungal DNA content per 10 ng, whereas the diagrams display percentages. Line 353, Figure S4: Please rephrase this sentence. Furthermore, linearity is difficult to appreciate. The authors should either include an additional diagram similar to Figure S3 or color-code the different inoculums. Line 363-367: Please clarify the denominator in the text, figure legend, and/or axis label. The methods section suggests “spores per 20 mg tissue” How was “consistency” determined? The figure does not contain an indication of variance/SD/SEM. Which coefficients of variation were seen in technical replicates and independent biological replicates? A lower limit of linearity at 10e4 spores per 20 mg tissue would not be particularly exciting for fungal burden determination in murine model. In addition, the slopes are not very consistent. How does the rnl marker perform compared with 18S in your organ models? Line 382-384: What is the rationale for this specific cutoff value? Are 95 % or 99 % confidence intervals for spores and germinated stages overlapping? Line 394-395: Why have the two modalities been performed in a different number/selection of tissue sections? Line 422-426: Mentioning these critical limitations in the Introduction would strengthen the authors’ outline of a critical need for novel molecular markers (Line 69-71, where it may not become fully clear for a non-expert reader why ribosomal markers are not sufficient).
Author Response
Response to the Reviewers
Reviewer 1
The difficult and often delayed diagnosis remains a major obstacles to reduce the mortality of mucormycosis. There is lack of biomarkers efficiently detecting Mucorales from a broad spectrum of clinical matrices and many of the existing markers lack the ability to reliably distinguish Mucorales from other pathogenic molds. Therefore, Caramalho and colleagues evaluated the potential of a novel mitochondrial molecular marker (rnl-qPCR assay). The study has been very thoroughly designed and executed and conclusions are supported by an impressive amount of validation experiments and supplementary data. The ability to distinguish resting and germinated growth stages as part of a molecular assay represents a significant advancement. Overall, the assay is of major interest for both translational/clinical mycology and pre-clinical studies in animal models. My only major concern refers to the relatively high limit of linearity in murine organ models (see comments below).
However, the manuscript would greatly benefit from careful revisions by a native speaker, as there is a multitude of grammatical errors and poorly constructed sentences (esp. in the Discussion section). Similarly, some of the figures are not very well designed and, at times, labels are difficult to read (e.g. in Figure 6). More consistent formatting, larger fonts, and a better resolution wound enhance the presentation of the figures.
Response: A native speaker has improved the manuscript to improve gramma and readability. Figures were overworked to improve readability.
Specific comments:
Multiple figures are numbered wrongly. Line 302: Table S1 does not provide any information regarding comparative outcomes of ITS and rnl assays.
Response: The reviewer is correct that should be Table S4
Lines: 345-348: Please carefully double-check both the text and figure. The diagrams do not seem to fully correspond with the ranking and numbers provided in the test.
Response: The numbers are accurate, but I guess the Figure was difficult to interpret; we improved that by adding two sentences to the figure legend.
In panel S3B, the limit of linearity remains unclear, since the end of the regression line and shaded area do not match.
Response: The linearity is indicated by the green shaded box, the regression trend presented was 0.9985 based on the values inside the green shaded box, as the regression is 0.9985 even when the lowest value was included we decide to include the value, but the reviewer is right one could argue that the linearity ends a dilution higher. We changed it according to your suggestion and modified the graphic and the text accordingly.
In addition, the numbers are difficult to appreciate, as the text refers to fungal DNA content per 10 ng, whereas the diagrams display percentages.
Response: We changed the title of the y-axis.
Line 353, Figure S4: Please rephrase this sentence. Furthermore, linearity is difficult to appreciate. The authors should either include an additional diagram similar to Figure S3 or color-code the different inoculums.
Response: We color-coded the different concentrations to improved readability.
Line 363-367: Please clarify the denominator in the text, figure legend, and/or axis label. The methods section suggests “spores per 20 mg tissue” How was “consistency” determined?
Response: We added the denominator to the text and also to the figure legend and in the axis to make the results clear. Consistencies were calculated based on the biological replicates.
The figure does not contain an indication of variance/SD/SEM. Which coefficients of variation were seen in technical replicates and independent biological replicates?
Response: Adding the SDs to the graphic would make the graphic very hard to read; we gave the SDs in a separate supplementary table S5.
A lower limit of linearity at 10e4 spores per 20 mg tissue would not be particularly exciting for fungal burden determination in murine model.
Response: We are sorry, but these are the values, we know that this sounds not striking having other fungal pathogens in mind, but for mucormycetes were DNA extraction is difficult it is comparable with what other have achieved. However we showed that the assay works of human FFPE so LOD for hyphae might be much lower, but hyphae are difficult to quantitatively spike into clinical specimens. Further studies of fresh tissue for patients will certainly give us a better idea of the LOD in real clinical specimens.
In addition, the slopes are not very consistent. How does the rnl marker perform compared with 18S in your organ models?
Response: We did not test 18S in comparison to rnl, but from the TEF data for the quantification you can see that for active stages mitochondrial markers outperform nucleic marker TEF. The 18 S PCR was only used for comparison in clinical specimens, as the assays are very expensive as this is a IVD-CE certified diagnostic kit.
Line 382-384: What is the rationale for this specific cutoff value? Are 95 % or 99 % confidence intervals for spores and germinated stages overlapping?
Response: The specific cut off value allows differentiating active from resting stages that is the rationale behind giving the cut off value. Active stages have more mitochondria than resisting stages. The fold changes basically allow you to state if a fungus is resting or active. The confidence intervals of resting and active stages do not overlap (see graphics), but between germlings and hyphae as both have high energy demands.
Line 394-395: Why have the two modalities been performed in a different number/selection of tissue sections?
Response: That was only depended on what has been performed in routine diagnostic and if clinical material was available to be studies, as this was a retrospective study. Evaluations on bigger sets of clinical specimens are needed for a follow up study.
Line 422-426: Mentioning these critical limitations in the Introduction would strengthen the authors’ outline of a critical need for novel molecular markers (Line 69-71, where it may not become fully clear for a non-expert reader why ribosomal markers are not sufficient).
Response: A paragraph was added to the introduction to make this clearer for a wider readership.
Reviewer 2 Report
Dear authors,
You have well evaluated and described Rnl as a Mucorales marker as well as as a species identification marker within this group. The study design was well developed to assess specificity and sensitivity bu using in vitro as well as in silico approaches.
I also appreciated very much the fact that it was also evaluated on murine organs and FFPE, underlining the potential of this approach in diagnostics.
I have only a few remarks to address:
Please check the numbering of the figures and tables and the references to them in the text. (e.g. in the Legend of Figure 2 it says Figure 1, For Figure 4 it says Figure 2 in the legend,...) Please verify and correct this. I would consider making Table S2 directly available and not as supplemental material In Figure 1: please be consequent with the taxonomy: replace 'Rhizopus oryae' with 'R. arrhizus' in the figure itself
One more remark is that I find the term pan-Mucorales marker confusing in that sense that it implies that it is identical in all Mucorales, which is not the case. Just explain this in the text please.
Best regards
Author Response
Reviewer 2
You have well evaluated and described Rnl as a Mucorales marker as well as as a species identification marker within this group. The study design was well developed to assess specificity and sensitivity bu using in vitro as well as in silico approaches.
Response: Thank you.
I also appreciated very much the fact that it was also evaluated on murine organs and FFPE, underlining the potential of this approach in diagnostics.
Response: Thank you.
I have only a few remarks to address:
Please check the numbering of the figures and tables and the references to them in the text. (e.g. in the Legend of Figure 2 it says Figure 1, For Figure 4 it says Figure 2 in the legend,...) Please verify and correct this.
Response: We rechecked the appearance of figures in the text and corrected order if needed.
I would consider making Table S2 directly available and not as supplemental material
Response: We see where the reviewer is coming from but as we have already so many Figures presented in the main manuscript, we prefer to leave the Table S2 in the Supplements do not further stretch the paper in length.
In Figure 1: please be consequent with the taxonomy: replace 'Rhizopus oryae' with 'R. arrhizus' in the figure itself
Response: Was changed.
One more remark is that I find the term pan-Mucorales marker confusing in that sense that it implies that it is identical in all Mucorales, which is not the case. Just explain this in the text please.
Response: We explained that better. The primers work for all mucormycetes and would give you positive amplicons, but the amplicons vary in size that allows us to distinguish the amplicons with HRM analysis.
Round 2
Reviewer 1 Report
I would like to thank the authors for their responsiveness to my and the other reviewer’s comments. The revisions have enhanced the scientific content and the clarity of the text and figures for the average reader. In particular, the manuscript strongly benefited from the author’s language editing and improvements of the figures.
However, the manuscript still contains several dozens of spelling and punctuation mistakes, grammar/syntax issues, word duplications, and style issues. While these could be mitigated in the editorial editing stage, I believe that the scientific depth and impact of the study would warrant another cycle of language editing by the authors, also aiming to improve the style and flow of the text.
In addition, there are a few minor points I would like to bring to the authors’ attention:
Line 344-348: While the updated figure is much easier to interpret, the numbers provided in the text for L. ramosa and the two Rhizopus species are still discrepant from Figure S4. For example, the R. arrhizus panel shows a range of linearity from 0.001 ng (= 1 pg) to 1 ng, whereas the text states 100 pg. For L. ramosa and R. microsporus, it should be 10 pg – 1 ng. For the latter two species, the authors state the upper limit of linearity (higher concentration, left margin in the figure), whereas the lower limit (lower concentration, right margin) is provided for all other species. Line 368-369: While acknowledging the technical challenge of DNA isolation from Mucorales spores, in our hands, the lower limit of detection for the 18S assay often reaches 10e4 spore equivalents per murine lung, equaling less than 10e3 spores per 20 mg tissue. The authors’ rnl assay also shows considerable variation of some of Ct values, and is, for several samples, nowhere close to the optimal 3-3.5 Ct delta for 10-fold serial dilutions. However, the many advantages of the authors’ novel rnl based method may clearly outweigh these issues. I would just want to suggest to slightly tone down the concluding statement in lines 368-369, as the comparative merits of the assays for pre-clinical studies in animal models remain to be defined. Figure legend 6 now states “values are given per g tissue”, whereas the Results text, Methods text, and reply to my comment suggest “per 20 mg”. Line 384-385: The authors state that “the results define a cut-off value of 5.60”. Their response to my previous comment does fully address the question how this specific cut-off value was determined. In Fig. 7, it appears that the cut-off is much closer to the “spores” sample population than germinated stages (germlings and hyphae). Although in this particular selection of samples, 5.60 was able to correctly discriminate all samples, I wondered whether the calculation of 95% or 99% confidence intervals would result in a higher cut-off yielding a minimal projected overlap of the distributions for future, independent samples of the same type. In general, I would assume that this cut-off value would need to be determined separately for each type of specimen, a fact that could be added to the Discussion section. There are a couple of minor issues in the Supplement, e.g. the table header of Table S2 should be placed on page 2 directly above the table, legend S4 has a typo in the species name L. ramosa, and Figure S5 has a wrong number.
Author Response
I would like to thank the authors for their responsiveness to my and the other reviewer’s comments. The revisions have enhanced the scientific content and the clarity of the text and figures for the average reader. In particular, the manuscript strongly benefited from the author’s language editing and improvements of the figures.
Thank you for this positive response to the efforts we made.
However, the manuscript still contains several dozens of spelling and punctuation mistakes, grammar/syntax issues, word duplications, and style issues. While these could be mitigated in the editorial editing stage, I believe that the scientific depth and impact of the study would warrant another cycle of language editing by the authors, also aiming to improve the style and flow of the text.
Sorry if language issues caused inconveniences. We had another thorough read through and asked again a native speaker to proof read the article again for text flow. We would be grateful if remaining language issues can be addressed by the copy editor.
In addition, there are a few minor points I would like to bring to the authors’ attention:
Line 344-348: While the updated figure is much easier to interpret, the numbers provided in the text for L. ramosa and the two Rhizopus species are still discrepant from Figure S4. For example, the R. arrhizus panel shows a range of linearity from 0.001 ng (= 1 pg) to 1 ng, whereas the text states 100 pg. For L. ramosa and R. microsporus, it should be 10 pg – 1 ng. For the latter two species, the authors state the upper limit of linearity (higher concentration, left margin in the figure), whereas the lower limit (lower concentration, right margin) is provided for all other species.
We completely rewrote the paragraph and are giving now the ranges of linearity for each species.
Line 368-369: While acknowledging the technical challenge of DNA isolation from Mucorales spores, in our hands, the lower limit of detection for the 18S assay often reaches 10e4 spore equivalents per murine lung, equaling less than 10e3 spores per 20 mg tissue. The authors’ rnl assay also shows considerable variation of some of Ct values, and is, for several samples, nowhere close to the optimal 3-3.5 Ct delta for 10-fold serial dilutions. However, the many advantages of the authors’ novel rnl based method may clearly outweigh these issues. I would just want to suggest to slightly tone down the concluding statement in
We changed it and it reads now as follows: “The consistent results delivered by rnl qPCR show rnl to be a robust marker for fungal burden evaluation (Table S5). Evaluation by other laboratories should now be a priority. “
lines 368-369, as the comparative merits of the assays for pre-clinical studies in animal models remain to be defined. Figure legend 6 now states “values are given per g tissue”, whereas the Results text, Methods text, and reply to my comment suggest “per 20 mg”.
The reviewer is right, there is still a mistake in the Figure 6 legend, we corrected it.
Line 384-385: The authors state that “the results define a cut-off value of 5.60”. Their response to my previous comment does fully address the question how this specific cut-off value was determined.
This cut off value was determined, based on the range of fold changes seen for rnl/tef and spores. No fold change higher was gained for resting spores, also taking SD into consideration, so 5.6 seemed to be a safe cut off value to distinguish between resting conidia and the active stages (germlings and hyphae). But one could argue on that cut off and could decide to choose any cut off between 5.6 and 6.9 fold change as this is the window between resisting spores and active conidia.
In Fig. 7, it appears that the cut-off is much closer to the “spores” sample population than germinated stages (germlings and hyphae). Although in this particular selection of samples, 5.60 was able to correctly discriminate all samples, I wondered whether the calculation of 95% or 99% confidence intervals would result in a higher cut-off yielding a minimal projected overlap of the distributions for future, independent samples of the same type. In general, I would assume that this cut-off value would need to be determined separately for each type of specimen, a fact that could be added to the Discussion section.
The aim was to see if we actually can clearly distinguish resting conidia from active stages so we introduced the cut off value based on the results gained. The reviewer is right. We agree that for future studies this cut off needs to be set for every fungal species tested. We do not believe that the specimen tested makes a huge difference as fold change just reflects on the higher energy demand needed for germination and hyphal growth therefore more mitochondria are present in the cytoplasm per cell core and the fold change changes. We do not believe that this is very variable for different media but more so by fungal species. We will add a paragraph to the discussion to make that clear.
Discussion now reads as follows:
Resting stages had a 2∆Ct < 5.60 and active stages had a 2∆Ct > 5.60. This value was found to discrimination between the growth stages for all mucormycetes tested: R. arrhizus, R. microsporus, M. circinelloides, and Lichtheimia spp. in our experiments. This cut off value might vary for different PCR-platforms, experimental conditions and fungal species tested. As this is the first study using this approach we have no comparative data from other laboratories using, this remain to be evaluated in future studies. The ability to discriminate between the various growth stages may also be of interest for other human-pathogenic fungi such as aspergilli in respiratory tract samples such as bronchoalveolar lavages, where discrimination between colonization and infection during aspergillosis is still a significant challenge. [61]
There are a couple of minor issues in the Supplement, e.g. the table header of Table S2 should be placed on page 2 directly above the table, legend S4 has a typo in the species name L. ramosa, and Figure S5 has a wrong number.
We overworked the supplementary tables and figures.